# Structure of amyloid-β (20-34) with Alzheimer's-associated isomerization at Asp23 reveals a distinct protofilament interface

Rebeccah A. Warmack [1], David R. Boyer[1], Chih-Te Zee[1], Logan S. Richards[1], Michael R. Sawaya [1,2,3], Duilio Cascio [1,2,3], Tamir Gonen [2,4,5,6], David S. Eisenberg [1,2,3,4,5] & Steven G. Clarke[1,2]

Amyloid-β (Aβ) harbors numerous posttranslational modifications (PTMs) that may affect Alzheimer's disease (AD) pathogenesis. Here we present the 1.1 Å resolution MicroED structure of an Aβ 20–34 fibril with and without the disease-associated PTM, L-isoaspartate, at position 23 (L-isoAsp23). Both wild-type and L-isoAsp23 protofilaments adopt β-helix-like folds with tightly packed cores, resembling the cores of full-length fibrillar Aβ structures, and both self-associate through two distinct interfaces. One of these is a unique Aβ interface strengthened by the isoaspartyl modification. Powder diffraction patterns suggest a similar structure may be adopted by protofilaments of an analogous segment containing the heritable Iowa mutation, Asp23Asn. Consistent with its early onset phenotype in patients, Asp23Asn accelerates aggregation of Aβ 20–34, as does the L-isoAsp23 modification. These structures suggest that the enhanced amyloidogenicity of the modified Aβ segments may also reduce the concentration required to achieve nucleation and therefore help spur the pathogenesis of AD.

[1] Department of Chemistry and Biochemistry, University of California, Los Angeles, Los Angeles, CA 90095-1569, USA. [2] Molecular Biology Institute, University of California, Los Angeles, Los Angeles, CA 90095-1570, USA. [3] UCLA-DOE Institute, University of California, Los Angeles, Los Angeles, CA 90095-1570, USA. [4] Howard Hughes Medical Institute, University of California, Los Angeles, Los Angeles, CA 90095-1570, USA. [5] Department of Biological Chemistry, University of California, Los Angeles, Los Angeles, CA 90095-1737, USA. [6] Department of Physiology, University of California, Los Angeles, Los Angeles, CA 90095-1751, USA. Correspondence and requests for materials should be addressed to S.G.C. (email: clarke@mbi.ucla.edu)

A prevalent theory for the biochemical basis of Alzheimer's disease (AD) is the amyloid cascade hypothesis, which describes the aggregation of the Aβ peptide into oligomeric or fibrous structures that then trigger the formation of neurotoxic tau neurofibrillary tangles[1–3]. The Aβ peptide is subject to a number of posttranslational modifications (PTMs) that may affect its aggregation in vivo[4]. Specifically, Aβ phosphorylation (Ser8, Ser26), pyroglutamylation (Glu3, Glu11), nitration (Tyr10), and racemization/isomerization (Asp1, Asp7, Asp23, Ser26) have been shown in vitro to increase the aggregation propensity or neurotoxicity of the Aβ 1–42 peptide[5–11], while other modifications, such as dityrosine crosslinking (Tyr10), have been shown to increase the stability of the Aβ aggregates[12].

Isomerized products of aspartic acid residues perturb protein structure by rerouting the peptide backbone through the side chain β-carbonyl. This age-dependent modification introduces a methylene group within the polypeptide backbone and thus may have a significant effect on the structure of Aβ oligomers or fibrils[13–15]. In addition, the isopeptide bond is resistant to degradation, potentially increasing the concentration of the isomerized Aβ form with respect to the native peptide. Despite the presence of a repair enzyme in the brain, the L-isoaspartate (D-aspartate) O-methyltransferase (PCMT1) for L-isoaspartate, the isomerization of Aβ Asp1, Asp7, and Asp23 has been identified within AD brain parenchyma[16,17]. In the cases of the heritable early-onset AD Iowa mutation (Asp23Asn), 25–65% of Asn23 residues have been shown to be isomerized in frontal lobe tissues[18], consistent with the increased rates of spontaneous deamidation/isomerization of asparagine relative to aspartate[19]. In vitro studies demonstrate that L-isoaspartate at Asp23 (L-isoAsp23) significantly accelerates Aβ 1–42 fibril formation, while L-isoAsp7 alone does not[11,20]. Subsequent studies using peptides with multiple sites of isomerization showed only minor accelerated aggregation of the tri-isomerized species (1, 7, and 23), over the di-isomerized species (7 and 23)[18]. Taken together, these results suggest that among the known sites of Asp isomerization in Aβ, L-isoAsp23 is primarily responsible for the increase in aggregation propensity in vitro.

Given the relevance of the isomerization of Asp23 to both sporadic and hereditary Iowa mutant forms of AD, we sought to discover the structural basis for its acceleration of fibril formation[10,17,18]. As a platform for evaluating this modification, we chose synthetically generated 15-mer peptides encoding residues 20–34 of the Aβ peptide (Aβ$^{20–34}$) with and without an L-isoAsp modification at position 23 and spanning the core of known Aβ fibril structures[21–29]. Challenged by the small size of crystals formed by this segment, we employed the cryo-electron microscopic (cryo-EM) method microcrystal electron diffraction (MicroED) to determine the structures. The structures of Aβ$^{20–34}$ and Aβ$^{20–34, isoAsp23}$, determined to 1.1 Å resolution by direct methods, reveal with atomic detail conserved kinked β-helix-like-turns with complex features similar to those observed previously at lower resolution in the cores of fibrillar Aβ 1–42, as well as a distinct pair of protofilament interfaces. Our results suggest that the L-isoAsp23 residue facilitates the formation of a more stable form of this unique interface, promoting enhanced fiber formation and stability. The length of these peptide segments, four residues longer than any other crystallographically determined amyloid structures[30–33], is key in facilitating their complex fold—a conformation more representative of the full-length Aβ fibrils.

## Results

### Fibril formation and characterization of Aβ$^{20–34}$ peptides.
Six early-onset hereditary Alzheimer's mutations and two PTMs, including the isomerized Asp23, are localized in the Aβ 1–42

peptide to a region spanning six residues from Ala21 to Ser26 near the center of the peptide (Fig. 1a, b)[17,34,35]. The amyloid-forming propensity of segments in this region of Aβ was assessed using a computational method of predicting steric zippers by a threading protocol (ZipperDB[36]). This method highlights a region of Aβ from Asn27 to Gly37 with high aggregation propensity near the site of Asp23 isomerization (Fig. 1b). To characterize segments containing an isomerized Asp residue at position 23, we utilized synthetic 15 residue peptides spanning the Aβ residues 20–34 (Aβ$^{20–34}$) in which Asp23 was substituted with either an L-Asn residue (Iowa mutant; Aβ$^{20–34, Asp23Asn}$) or an L-isoAsp residue (Aβ$^{20–34, isoAsp23}$).

To evaluate the effect of these variations on this 15-residue segment of Aβ, we assayed its capacity to form fibrils as measured by light scattering at 340 nm (Fig. 1c). Both the peptide based on the Iowa mutant (Aβ$^{20–34, Asp23Asn}$) and the peptide based on L-isoAsp23 (Aβ$^{20–34, isoAsp23}$) demonstrated significantly enhanced fibril formation over that of Aβ$^{20–34}$, with the Iowa mutant peptide displaying the fastest initial rate of fibril formation (Fig. 1c). Fibers of the native peptide at this concentration (1.6 mM) were not observed by light scattering or EM. We further discovered that only 34% of these Aβ$^{20–34, isoAsp23}$ aggregates could be methylated by the L-isoAsp repair protein carboxyl methyltransferase (PCMT1) in vitro (Supplementary Fig. 1). These data suggest that a majority of the L-isoAsp sites are occluded from the normal repair pathway once in this aggregate form.

To determine the ability of these modified forms to accelerate the aggregation of native peptide, seeding of 3.2 mM Aβ$^{20–34}$ was performed using 10 μM final concentrations of pre-aggregated seeds of Aβ$^{20–34}$, Aβ$^{20–34, Asp23Asn}$, and Aβ$^{20–34, isoAsp23}$ (Fig. 1d). The addition of each of the preformed aggregates caused significant acceleration in the onset of fiber formation. The largest shift occurred with the native Aβ$^{20–34}$ seed, followed by isomerized Aβ$^{20–34, isoAsp23}$ and Aβ$^{20–34, Asp23Asn}$. Powder diffraction performed on the final aggregates revealed nearly identical sets of reflections, suggesting that the three seeds have similar enough structures to template wild-type (WT) Aβ$^{20–34}$ aggregates whose diffraction resembles unseeded fibrils (Supplementary Fig. 2a). Fibrillization experiments of full-length Aβ 1–40 with and without the L-isoAsp modification at residue 23 reveal that the isomerized species displays a shorter lag time, consistent with the results obtained with the corresponding Aβ$^{20–34}$ peptides (Supplementary Fig. 2b; Fig. 1c). Thus, while the isomerized form may be only a minor component of the in vivo Aβ population, it aggregates at a faster rate and can cross-seed the native form efficiently in vitro.

In contrast to the results obtained with 1.6 mM Aβ$^{20–34}$, increasing the concentration to 3.2 mM Aβ$^{20–34}$ did yield aggregates ~77 nm in width (Fig. 1c, d). Importantly, light scattering under these conditions for this native peptide is not detected until 3.5 h at the earliest, while shifts in light scattering for the 1.6 mM isomerized and mutated peptides were detected by 1.5 and 0.5 h, respectively (Fig. 1d). Direct comparisons of formation rates were complicated by the insolubility of the Aβ$^{20–34, Asp23Asn}$ peptide at high concentrations, but the delayed onset of even the 3.2 mM Aβ$^{20–34}$ incubation compared to the 1.6 mM Aβ$^{20–34, Asp23Asn}$ and Aβ$^{20–34, isoAsp23}$ incubations also support the increased rates of aggregation of the mutated and isomerized peptides (Fig. 1c, d).

Fibrils of each segment were also investigated for their resistance to dissociation by dilution into increasing concentrations of sodium dodecyl sulfate (SDS) at 70 °C as measured by light scattering at 340 nm (Fig. 2). Fibrils of the native 15-residue Aβ segment appeared to partially dissolve upon dilution into the SDS-free buffer, although remaining aggregates were found by

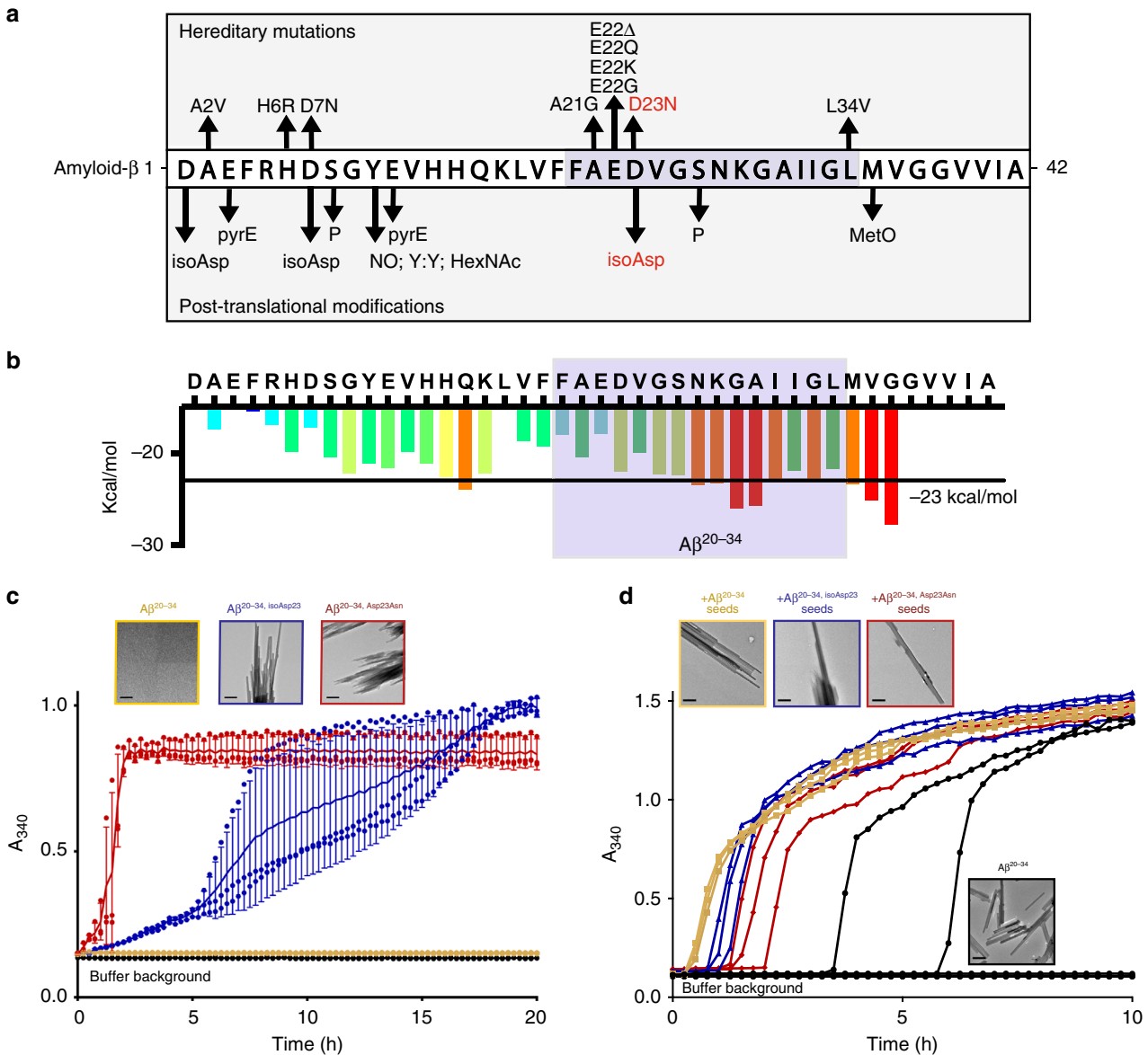

**Fig. 1** L-isoAsp in Aβ[20–34] accelerates fiber formation and can seed native segment. **a** Sequence of human Aβ including known early-onset hereditary mutations and posttranslational modifications (pyrE pyroglutamate, P phosphorylation, NO nitration, Y:Y dityrosine crosslink, HexNAc glycosylation, MetO oxidation). **b** ZipperDB[22] amyloid propensity profile for the human Aβ sequence with the Aβ[20–34] sequence highlighted in light blue. **c** 1.6 mM of Aβ[20–34], Aβ[20–34, Asp23Asn], and Aβ[20–34, isoAsp23] peptide aggregation was monitored by turbidity at 340 nm. Each data point is shown as a round symbol, the solid line represents the mean value, and error bars represent SD of three replicates. Transmission electron micrographs of aggregates are shown at the top left of the graph, scale bars represent 0.5 μm in each image. **d** Aggregation of 3.2 mM Aβ[20–34] in 50 mM Tris, pH 7.5, 150 mM NaCl, and 1% dimethyl sulfoxide was monitored by turbidity at 340 nm alone (black lines) or with 10 μM pre-aggregated seeds of Aβ[20–34] (yellow), Aβ[20–34, isoAsp23] (blue), and Aβ[20–34, Asp23Asn] (red). Each line represents a replicate well. Transmission electron micrographs of aggregates are shown at the top left of the graph; scale bars shown at the lower left represent 0.5 μm in each image. Source data are provided as a Source Data file

EM. However, these were completely dissolved upon incubation with 1% SDS and higher concentrations (Fig. 2). In contrast, the isomerized peptide showed increased resistance to dissolution compared to the native peptide and still showed light scattering at a concentration of 2% SDS, though no more aggregates were seen at 5% SDS (Fig. 2b). The fibrils of the Iowa mutant appeared to be largely unaffected by dilution even at the highest concentrations of SDS, with no significant changes observed in the levels of light scattering. However, the aggregates in 5% SDS seen by EM appeared to be less bundled than at lower concentrations (Fig. 2b). These results show that alterations of the structure at Asp23 strongly contribute to fibril formation and stability.

**Crystallization and data collection of the Aβ[20–34] segments**. To understand the atomic structural basis for changes in the properties of the isomerized peptide, we sought to crystallize it in the amyloid state. Vapor diffusion screening yielded no crystals large enough for analysis by conventional X-ray crystallography for either the Aβ[20–34] or the Aβ[20–34, isoAsp23] segment. Instead ordered nanocrystals of the native segment were obtained with continuous shaking at 1200 rpm, and ordered nanocrystals of the isomerized segment were generated with constant mixing using an acoustic resonant shaker[37,38] for analysis by MicroED[39,40] as described in the "Methods" section. Nanocrystals obtained in varying buffer conditions were evaluated by morphology and diffraction via light and EM, respectively. Those formed under

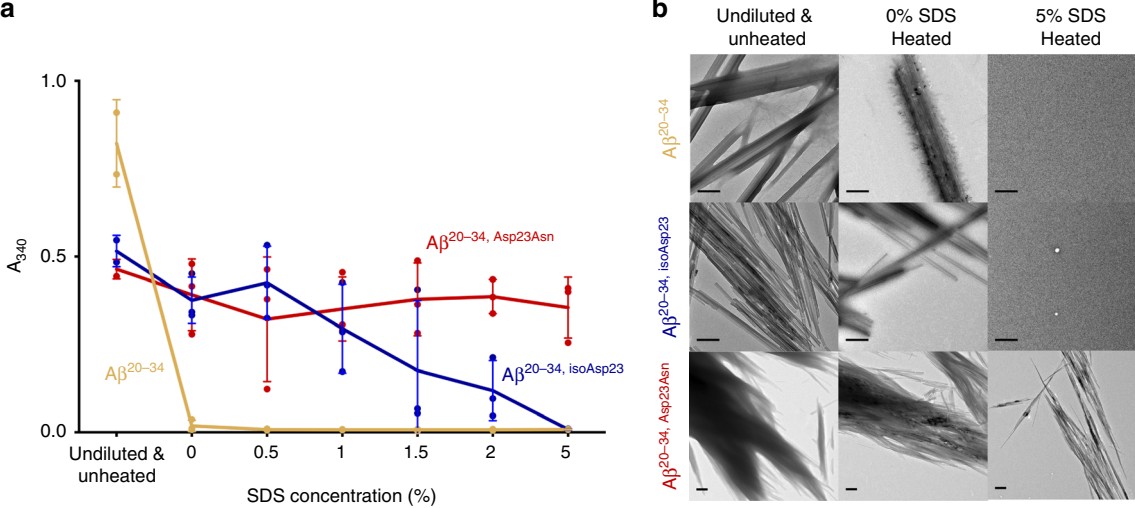

**Fig. 2** Modified fibers have increased resistance to sodium dodecyl sulfate (SDS) disaggregation. **a** Fiber stocks (Undiluted & Unheated initial points are two readings of the fiber stocks) were mixed 1:1 in buffer (0% SDS final) and increasing concentrations of SDS (1, 1.5, 2, 5% final) as described in the "Methods" section. Each data point is shown as a round symbol, the solid line represents the mean value, and error bars represent the SD of three technical replicates. **b** Transmission electron micrographs of disaggregated fibers, scale bars in the lower left represent 0.5 μm. Source data are provided as a Source Data file

the most promising conditions were used as seeds for additional rounds of batch crystal formation. The optimal crystallization condition for the isomerized segment was 50 mM Tris, pH 7.6, 150 mM NaCl, and 1% dimethyl sulfoxide (DMSO) for 48 h with 2% seeds. Crystals of the native segment grew in 50 mM Tris, pH 7.5, 150 mM NaCl, and 1% DMSO for 30 h without seeding. Isomerized crystal trials produced densely bundled nanocrystals that could not be disaggregated by sonication and freeze–thawing. However, washing crystal solutions with a 0.75% (w/v) solution of β-octyl glucoside in TBS, pH 7.6 yielded a higher number of single crystals for subsequent data collection. Dilution one to one in buffer yielded sufficient single crystals of the native segment for data collection (Fig. 3a, d). Data were collected on a Thermo Fisher TALOS Arctica microscope operating at 200 kV using a bottom mount CetaD CMOS detector. Each Aβ[20–34] nanocrystal could be rotated continuously up to 140 degrees during data collection. A 1.1-Å-resolution structure was obtained by direct methods for each segment as described in the "Methods" section; refinement statistics for the structures are shown in Table 1.

**MicroED structures of Aβ[20–34] and Aβ[20–34, isoAsp23] segments.** The structures of both the Aβ[20–34] and the Aβ[20–34,isoAsp23] protofilaments reveal parallel, in-register architectures in which individual peptide chains stack through backbone hydrogen bonds every 4.8 and 4.9 Å along the protofilament axis, respectively (Fig. 3). In cross-section, both protofilaments appear triangular owing to sharp turns (β-arches) at Gly25 and Gly29, which divide each chain into three short, straight segments (Fig. 3b, e and Supplementary Fig. 3a). When compared with the structures in the protein databank, the three-sided Aβ[20–34,iso-Asp23] structure aligns best with a β-helical antifreeze protein from *Marinomonas primoryensis* but lacks linker regions between each stacked chain. We thus designate this amyloid motif as a β-helix-like turn (Supplementary Fig. 4)[41,42]. At the central core of both Aβ[20–34] and the Aβ[20–34,isoAsp23] protofilaments are the buried side chains of Phe20, Ala21, Val24, Asn27, and Ile31 in a zipper-like "intraface" that is completely dry. The side chain of Asn27 further stabilizes the assembly by forming a ladder of hydrogen bonds (polar zipper) along the length of the protofilament[43] (Supplementary Fig. 3b).

Each protofilament self-associates with neighboring protofilaments in the crystals through two distinct interfaces. Interface A in both structures resembles a canonical steric zipper—with intersheet distances of 8.3 and 9.1 Å for the native and isomerized, respectively (Fig. 3b, e). Both are lined by the hydrophobic side chains of Ala30, Ile32, and Leu34 that are related by 2₁ screw symmetry (steric zipper symmetry class 1[44]). Interface A is completely dry owing to a high $S_c$ of 0.73 in the native and 0.62 in the isomerized. This interface buries approximately 130 Å² per chain in the native form and 131 Å² in the isomerized form.

Unlike the dry steric zipper interface A, six water molecules line the second Aβ[20–34] interface, which we designate the "L-Asp Interface B" (Fig. 3b, e). Here the protofilaments are also related by a two-fold screw symmetry axis. Nearest this central axis, Gly25 and Ser26 contact their symmetry partners across the interface, separated by only 3.5 Å. Furthest from the axis, Asp23 and Lys28 from opposing protofilaments form charged pairs. In between each of these two regions is a solvent channel with the three ordered waters, yielding low shape complementarity ($S_c =$ 0.43) to this interface overall. In contrast, in the Aβ[20–34,isoAsp23] "L-isoAsp interface B" the truncated side chain of the L-isoAsp23 residue no longer forms a charged pair with Lys28 and instead the isomerized protofilaments form a completely dry interface containing the methylene group of L-isoAsp23, Val24, Gly25, and Ser26 with high surface complementarity ($S_c = 0.81$; Fig. 3e). This interface is tightly mated over its entire surface with an average distance of 4.0 Å between the backbones. Interface B buries approximately 139 and 122 Å² per chain for the native and isomerized forms, respectively. The exclusion of water molecules from the L-isoAsp interface B likely results in a favorable gain in entropy for the structure, and there are attractive van der Waals forces along the tightly mated residues L-isoAsp23-Ser26.

**Powder diffraction studies of Aβ peptides.** X-ray powder diffraction (XRD) patterns revealed that the fibrils of Aβ[20–34] segments appear largely isomorphous, sharing major reflections at ~4.7, 10, 12.2, 14, and 29–31 Å (Fig. 4a, b). The similarity among the powder diffraction patterns of Aβ[20–34, isoAsp23], Aβ[20–34, Asp23Asn], and Aβ[20–34] indicates that Aβ[20–34, Asp23Asn] mimics the

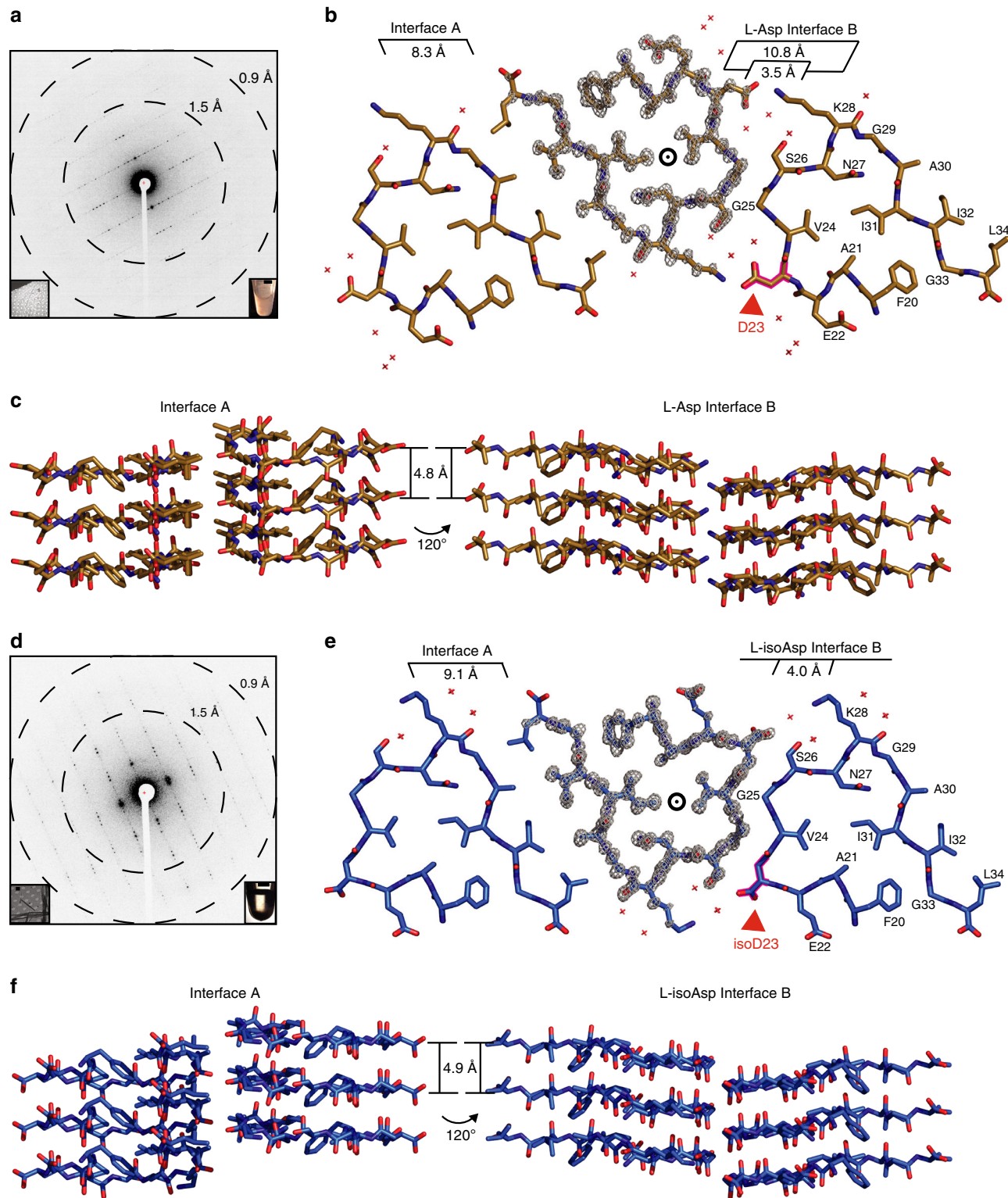

structures of the native and isomerized segments. We modeled an L-Asn residue at position 23 of the Aβ$^{20-34,\ isoAsp23}$ structure to see if the native L-amino acid could be accommodated in the dry L-isoAsp interface B (Fig. 4c, right panel). The L-Asn residue was integrated into the Aβ$^{20-34,\ isoAsp23}$ interface B scaffold without significant clashes. However, this Asn model lacks a backbone hydrogen bond extending between the isoAsp23 amide carboxyl to the Val24 amide nitrogen of the adjacent protofilament that is

present in our Aβ$^{20-34,\ isoAsp23}$ structure (Fig. 4c). The residue at site 23 has to adopt an allowed, but unusual left-handed helical conformation to form the L-isoAsp interface B. Both the methylene of the isoAsp residue and the isoAsp23 to Val24 main chain hydrogen bond may help stabilize this structure. This backbone hydrogen bond is present in the native Aβ$^{20-34}$ structure (Fig. 4c). In this native structure, the Asp main chain adopts a more canonical β-sheet conformation, but the side chain

**Fig. 3** $A\beta^{20-34, \, isoAsp23}$ structure contains an altered protofilament interface. **a** Representative single crystal electron diffraction pattern of $A\beta^{20-34}$ with resolution rings obtained during microcrystal electron diffraction (MicroED) data collection. Left inset shows the diffracting crystal (lower left scale bar represents 1 μm). Right inset shows light microscopic image of microcrystal sediment in a 1.6-mL microfuge tube (scale bar represents 3 mm). **b** One layer of the $A\beta^{20-34}$ crystal structure viewed down the fibril axis highlighting two distinct steric zipper interfaces. Interface distances are labeled (center circle indicates fibril axis). Waters are represented by red crosses. The $2F_o - F_c$ density is shown as a gray mesh at $2\sigma$ on the center protofilament. The Asp23 residue is outlined in magenta and shown by the red arrow. **c** Three layers of the $A\beta^{20-34}$ structure viewed perpendicular to the fibril axis (indicated by arrows; Left panel—interface A, right panel–L-Asp interface B). **d** Representative single crystal electron diffraction pattern of $A\beta^{20-34, \, isoAsp23}$ with resolution rings obtained during MicroED data collection. Left inset shows the diffracting crystal (lower left scale bar represents 1 μm). Right inset shows a light microscopic image of microcrystal sediment in a 1.6-mL microfuge tube (scale bar represents 3 mm). **e** One layer of the $A\beta^{20-34, \, isoAsp23}$ crystal structure viewed down the fibril axis highlighting two distinct steric zipper interfaces. Interface distances are labeled (center circle indicates fibril axis). Waters are represented by red crosses. The $2F_o - F_c$ density is shown as a gray mesh at $2\sigma$ on the center protofilament. The L-isoAsp23 residue is outlined in magenta and shown by the red arrow. **f**, Three layers of the $A\beta^{20-34, \, isoAsp23}$ structure viewed perpendicular to the fibril axis (indicated by arrows; Left panel—interface A, right panel—L-isoAsp interface B)

| Table 1 Data collection and refinement statistics | | |
|---|---|---|
| | $A\beta^{20-34}$ | $A\beta^{20-34, \, isoAsp23}$ |
| *Data collection* | | |
| Space group | P2$_1$ | P2$_1$ |
| Cell dimensions | | |
| *a, b, c* (Å) | 33.17, 4.78, 30.33 | 29.20, 4.87, 32.44 |
| *α, β, γ* (°) | 90.00, 111.10, 90.00 | 90.00, 101.90, 90.00 |
| Resolution (Å) | 1.10 (1.13–1.10)$^a$ | 1.05 (1.20–1.05)$^{b,c}$ |
| $R_{sym}$ or $R_{merge}$ (%) | 18.9 | 19.7 |
| $I/\sigma I$ | 5.41 (3.28) | 3.76 (1.38) |
| Completeness (%) | 85.2 | 82.7 (53.0) |
| Redundancy | 6.67 (6.14) | 4.19 (3.10) |
| *Refinement* | | |
| Resolution (Å) | 7.74–1.10 (1.26–1.10) | 5.96–1.05 (1.20–1.05) |
| No. of reflections | 3544 (1141) | 3943 (1167) |
| $R_{work}/R_{free}$ (%) | 19.4/21.3 (21.3/26.9) | 19.7/24.6 (27.0/32.4) |
| No. of atoms | | |
| Protein | 210 | 204 |
| Ligand/ion | 0 | 0 |
| Water | 7 | 4 |
| *B-factors* | | |
| Protein | 6.50 | 8.29 |
| Ligand/ion | — | — |
| Water | 20.78 | 27.70 |
| *R.m.s. deviations* | | |
| Bond lengths (Å) | 0.56 | 1.04 |
| Bond angles (°) | 0.68 | 0.90 |

$^a$Ten crystals were used in determining the $A\beta^{20-34}$ structure
$^b$Five crystals were used in determining the $A\beta^{20-34, \, isoAsp23}$ structure
$^c$Values in parentheses are for the highest-resolution shell

protrudes toward the opposite protofilament, prohibiting a tight, dry interface along residues Asp23-Ser26 as in the L-isoAsp interface B.

The L-Asn side chain in the L-isoAsp interface B model may be able to compensate for the loss of this dry interface by forming another ladder of hydrogen bonds along the protofilament axis (Fig. 4c, d right panel). Thus this second interface packing may be achievable for a $A\beta^{20-34, \, Asp23Asn}$ structure as shown in the L-isoAsp interface B model; however, the XRD patterns reveal that the native $A\beta^{20-34}$ and mutated $A\beta^{20-34, \, Asp23Asn}$ peptides share more similarities than the isomerized $A\beta^{20-34, \, isoAsp23}$ and the $A\beta^{20-34, \, Asp23Asn}$ peptide. Both the native and heritable Iowa mutant forms lack more defined peaks at 22.9, 24.7, 29.4, and 32.5, while both have more broad peaks at 30.9 Å (Fig. 4a, b). These similarities between the $A\beta^{20-34}$ and $A\beta^{20-34, \, Asp23Asn}$ fiber diffractions patterns, and the lack of a methylene group in the normal L-residues, may suggest that the Iowa mutant

$A\beta^{20-34, \, Asp23Asn}$ peptide will assume a structure more similar to the native $A\beta^{20-34}$ structure, as modeled in Fig. 4c, d (left panels). This model maintains the backbone hydrogen bond between Asn23 and Val24, the ordered core of the $A\beta^{20-34}$ structure, and allows for the additional polar zipper between stacked Asn23 residues. The added network of hydrogen bonds along the asparagine side chain may explain in part the increased fiber formation rates and stability of $A\beta^{20-34, \, Asp23Asn}$ against SDS and heat denaturation. While the isomorphous powder diffraction patterns seen between $A\beta^{20-34, \, isoAsp23}$, $A\beta^{20-34, \, Asp23Asn}$, and $A\beta^{20-34}$ do support the models in which $A\beta^{20-34, \, Asp23Asn}$ mimics the native and isomerized structures, it cannot be ruled out that $A\beta^{20-34, \, Asp23Asn}$ forms a distinct structure, perhaps lacking either the L-Asp or the L-isoAsp novel interface B, with the ordered core simply stabilized further by the Asn polar ladder.

Importantly, the powder diffraction of full-length Aβ and the shorter peptide segments all display cross-β patterns with strong reflections at ~4.7 and 9–10 Å (Fig. 4a), and the crystal structures of $A\beta^{20-34}$ and $A\beta^{20-34, \, isoAsp23}$ form parallel, in-register beta-sheets similar to other full-length Aβ structures. Thus we hypothesized that the $A\beta^{20-34, \, isoAsp23}$ structure could form the core of a distinct isomerized Aβ polymorph. To visualize a potential full-length fiber with the $A\beta^{20-34, \, isoAsp23}$ structure as its core, we added the remaining residues of Aβ 1–42 onto the ends of the $A\beta^{20-34, \, isoAsp23}$ protofilaments and energy minimized the entire model as described in the "Methods" section. The resulting model demonstrates that the remainder of the residues of Aβ 1–42 can be accommodated in a favorable conformation with the isomerized segment as a core with interface A or B as the primary interface (Fig. 5).

**Comparison of segment structures to known Aβ structures.** The structures presented here are the longest segments of an amyloid peptide determined by crystallography—four residues longer than the previous amyloid spines determined by MicroED[30–33]. This extension is significant due to the fact that, as the number of residues in a segment grows, the packing of idealized β-strands in a lattice becomes more difficult owing to the strain created by the natural twist of the β-sheet/strand. This strain hypothesis is consistent with observations that, as the number of residues in an amyloid segment grows, the crystals that can be grown are correspondingly smaller[45]. In the literature to date, the crystal structures of shorter segments of amyloid proteins have revealed that the dominant forces stabilizing proto-filaments occur between different peptide chains[46]. In the native and modified $A\beta^{20-34}$ structures, we are not only able to see interactions between protofilaments, such as the interfaces A and B, but we also see folding of the peptide to produce a β-helix-like

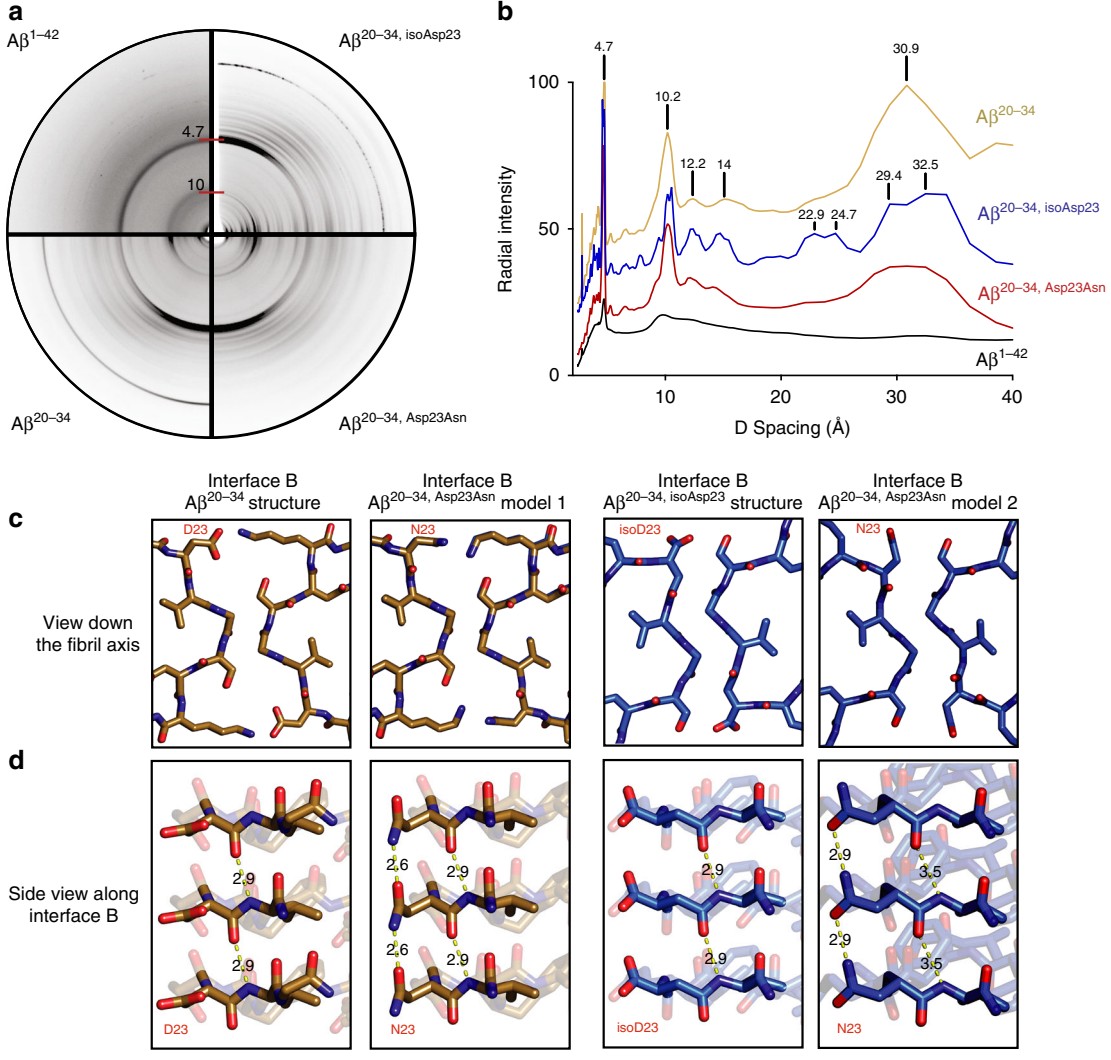

**Fig. 4** A putative model of the heritable Iowa mutation in interface B. **a** Fiber diffraction patterns of Aβ 1–42, Aβ$^{20-34}$, Aβ$^{20-34, \text{isoAsp23}}$, and Aβ$^{20-34, \text{Asp23Asn}}$. All fibers including Aβ 1–42 were prepared in 50 mM Tris, pH 7.6, 150 mM NaCl, and 1% dimethyl sulfoxide (DMSO), except Aβ$^{20-34, \text{Asp23Asn}}$, in which the DMSO concentration was raised to 5%. **b** Intensities of reflections from fiber diffraction of the segments were plotted against D spacing. Radial intensity values are vertically staggered for visibility of peaks. **c** From left to right, interface B down the fibril axis of Aβ$^{20-34}$ structure, a model Aβ$^{20-34, \text{Asp23Asn}}$ on the backbone of the Aβ$^{20-34}$ structure, Aβ$^{20-34, \text{isoAsp23}}$ structure, and a model Aβ$^{20-34, \text{Asp23Asn}}$ on the backbone of the Aβ$^{20-34, \text{isoAsp23}}$ structure. **d** A view perpendicular to the fibril axis of residues 23–24 of each structure. Yellow dashed lines represent measured distances in Å between the amide carboxyl of residue 23 and the amide nitrogen of Val24 on the adjacent strand. Source data are provided as a Source Data file

turn with a hydrophobic core of interacting residues within the same chain.

While not all full-length native structures contain β-arches, such as the peptide dimer structure shown in Schmidt et al.[47] (PDB code: 5AEF), all do include ordered cores involving steric zippers similar to those found in shorter amyloid peptide structures, and a majority of the known Aβ structures do display β-helix-like turns as seen in the segment structures (Fig. 6 and Supplementary Fig. 5). The native Aβ$^{20-34}$ structure aligns well with a number of these full-length Aβ structures, and both the native and isomerized structures presented here have the lowest total atom root-mean-square deviation (RMSD) with a structure of the Aβ Osaka mutant[29], E22Δ, at 2.741 and 2.963 Å, respectively. A tree representing the structural relationships between residues 20 and 34 of eight full-length Aβ structures and our Aβ$^{20-34}$ structure based on total atom RMSD values shows that 6 of the 8 structures contain turns about the Gly25 and Gly29 residues[21,23,24,26,28,29], creating interfaces which align well with

interface B of our L-Asp Aβ$^{20-34}$ structure. Four[21,23,26,29] of these structures correspond to both the Aβ$^{20-34}$ segment structures with regard to the placement of charged residues Glu22, Asp23, and Lys28 outside the hydrophobic core and yield total atom RMSD values of ≤4 Å with Aβ$^{20-34}$ (Fig. 6 and Supplementary Fig. 5). These strong overlaps between our segment structure and other full-length Aβ structures support the validity of this segment as an atomic resolution structure of an Aβ core. Importantly, in each of the full-length structures shown here, the putative interface B is accessible as a possible secondary nucleation site (Fig. 6). This interface is stabilized within our structures by the L-isoAsp modification, which mates more tightly between protofilaments than the L-Asp interface B and excludes waters. Thus a full-length structural polymorph with this interface may be isolated more readily with the modification.

The increased structural complexity afforded by extending from 11 to 15 residues is appreciated best in comparing the crystal structures of Aβ$^{20-34}$ to the shorter Aβ 24–34 crystal

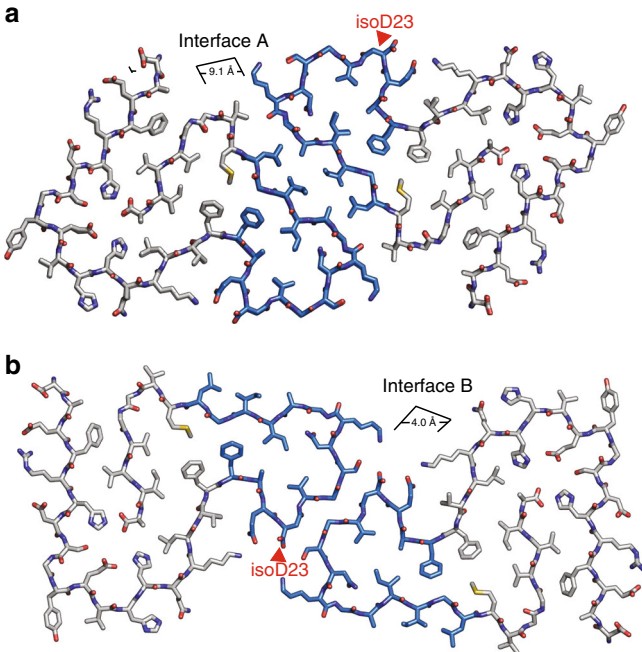

**Fig. 5** Model of Aβ[20–34, isoAsp23] as the core of an Aβ 1–42 modified polymorph. **a** Model of Aβ[1–42, isoAsp23] centered on interface A. **b** Centered on interface B. Blue residues correspond to the crystal structure core (Aβ[20–34, isoAsp23]), gray sticks correspond to the modeled extension (1–19 and 35–42)

structure, 5VOS[32] (Fig. 7). The four extra N-terminal residues in both native and modified Aβ[20–34] facilitate formation of kinks at Gly25 and Gly29, creating an internal core, whereas the Aβ 24–34 peptide assumes a linear β-strand. Despite Aβ 24–34 lacking these kinks, there is remarkable alignment between residues Gly29 to Leu 34 and interface A of the Aβ[20–34] crystals, yielding a total atom RMSD of 0.70 and 0.68 Å with the native and isomerized forms, respectively (Fig. 7). An inhibitor was previously developed to the human islet amyloid polypeptide (hIAPP) steric zipper interface analogous to this interface of the 5VOS Aβ 24–34 segment and was shown to be effective against fibril formation of both hIAPP and full-length Aβ[32]. Given the striking alignment between our Aβ[20–34] interface A and the 5VOS Gly29-Leu34 segment, as well as the distinct lack of modifications and mutations in the region of Asn27-Gly33, this interface may be an ideal scaffold for Aβ inhibitor design in both its homotypic steric zipper form as shown here or in the heterotypic zippers displayed in many of the full-length Aβ structures (Fig. 6).

## Discussion

The typical age of onset for sporadic AD is after 65 years, suggesting that slow spontaneous processes such as the accumulation of age-dependent PTMs in Aβ may be contributing factors to aggregation and toxicity[4]. The spontaneous isomerization of aspartate (isoAsp) has been identified at all three aspartate residues within the Aβ 1–42 peptide—1, 7, and 23. However, immunohistochemical studies have shown that, while native Aβ and isoAsp7 Aβ are present in senile plaques from four non-disease patient controls, isoAsp23 Aβ was identified only in one of the four non-disease patient controls, as well as in the senile plaques from all AD patient samples, indicating that the isoAsp23 may be more specifically associated with AD pathology than native Aβ and the L-isoAsp7 form[10]. This implied pathogenicity of isoAsp23 correlates with in vitro studies, which have

demonstrated accelerated amyloid formation of the isoAsp23 Aβ 1–40 and 1–42 peptides compared to native Aβ[10,11,17,18,20]. These results suggest that the change in the structure of Aβ accompanying isomerization at Asp23 may represent a route to the pathogenesis of AD.

In this work we present the 1.1 Å structures of segments spanning residues 20–34 of the Aβ peptide containing either an Asp or an isoAsp residue at site 23. These 15-residue segments, crystallized at physiological pH, maintain a topology seen in the core of Aβ fibrils, a β-helix-like turn (Fig. 6). The length of these peptides facilitates their similar overall fold to previous WT Aβ fibril structures and demonstrates that amyloid cores are rigid and ordered enough to form crystals. These structures reveal a previously unseen protofilament interface (B) involving residues Asp23-Lys28 in the native structure, and residues L-isoAsp23-Ser26 in the isomerized structure. The native interface (L-Asp interface B) has low surface complementarity and contains six water molecules encased between charged residue pairs Asp23 and Lys28 on opposing sheets. In contrast, the isomerized interface (L-isoAsp interface B) is a dry tightly mated sheet with high surface complementarity. Our data suggest that the changes in the structure along this interface, namely, the exclusion of water molecules and van der Waals attractive forces associated with the high $S_c$, are likely responsible in part for the increases in fiber formation rate and stability of the aggregate observed for the isomerized peptide. The modified interface may provide a better site for secondary nucleation of amyloid formation resulting in the observed enhancements in aggregation. However, it cannot be ruled out from the data presented here that the flexibility imparted by the methylene group of the L-isoAsp residue promotes amyloid formation by allowing an ordered nucleus for primary nucleation to form at a faster rate than the native peptide.

Our models of Asn23 in the Aβ[20–34, isoAsp23] "L-isoAsp interface B" indicate that the completely dry interface may be possible for native residues (Fig. 4). However, the native Aβ[20–34] structure did not preferentially adopt this interface and instead forms a hydrated L-Asp Interface B. Similar to our Aβ[20–34] peptide structure, alignments of previous Aβ structures onto the Aβ[20–34] and Aβ[20–34, isoAsp23] protofilaments show the native Asp23 side chain carboxyl group protruding into the putative interface B region (Supplementary Fig. 5). The hereditary Iowa mutant nuclear magnetic resonance (NMR) structure (Fig. 6 and Supplementary Fig. 5 (PDB: 2MPZ[22])) kinks at Gly25 and Asn27, rather than at Gly25 and Gly29, and thus there is no equivalent interface A. Yet, our preparations of crystals in TBS of the Aβ[20–34], Aβ[20–34, isoAsp23], and Aβ[20–34, Asp23Asn] constructs appear nearly identical by XRD, suggesting that the structure of an Iowa mutant protofilament would resemble the native and isomerized structures presented here (Fig. 4), barring minor differences due to packing polymorphisms or different environmental conditions.

It is clear that both the isomerization and Iowa mutation at residue 23 accelerate aggregation and increase stability of Aβ fibrils. Our structures of Aβ[20–34] and Aβ[20–34, isoAsp23] reveal a potential mechanism for the increases in fiber formation rate and fiber stability within the isoAsp23 form: the addition of a completely dry interface with high surface complementarity. This analysis leads to the hypothesis that the Asp23 isomerization in vivo could lead to the accelerated formation of Aβ fibrils, thereby contributing to the aggregation of Aβ and AD pathology. The hereditary Iowa mutation Asp23Asn may work in a similar manner either by forming the same fold as the isomerized Asp23 or, since Asn undergoes isomerization more rapidly relative to Asp, may also produce an isomerized Aβ with accelerated aggregation and increased stability. The isomerized structure may

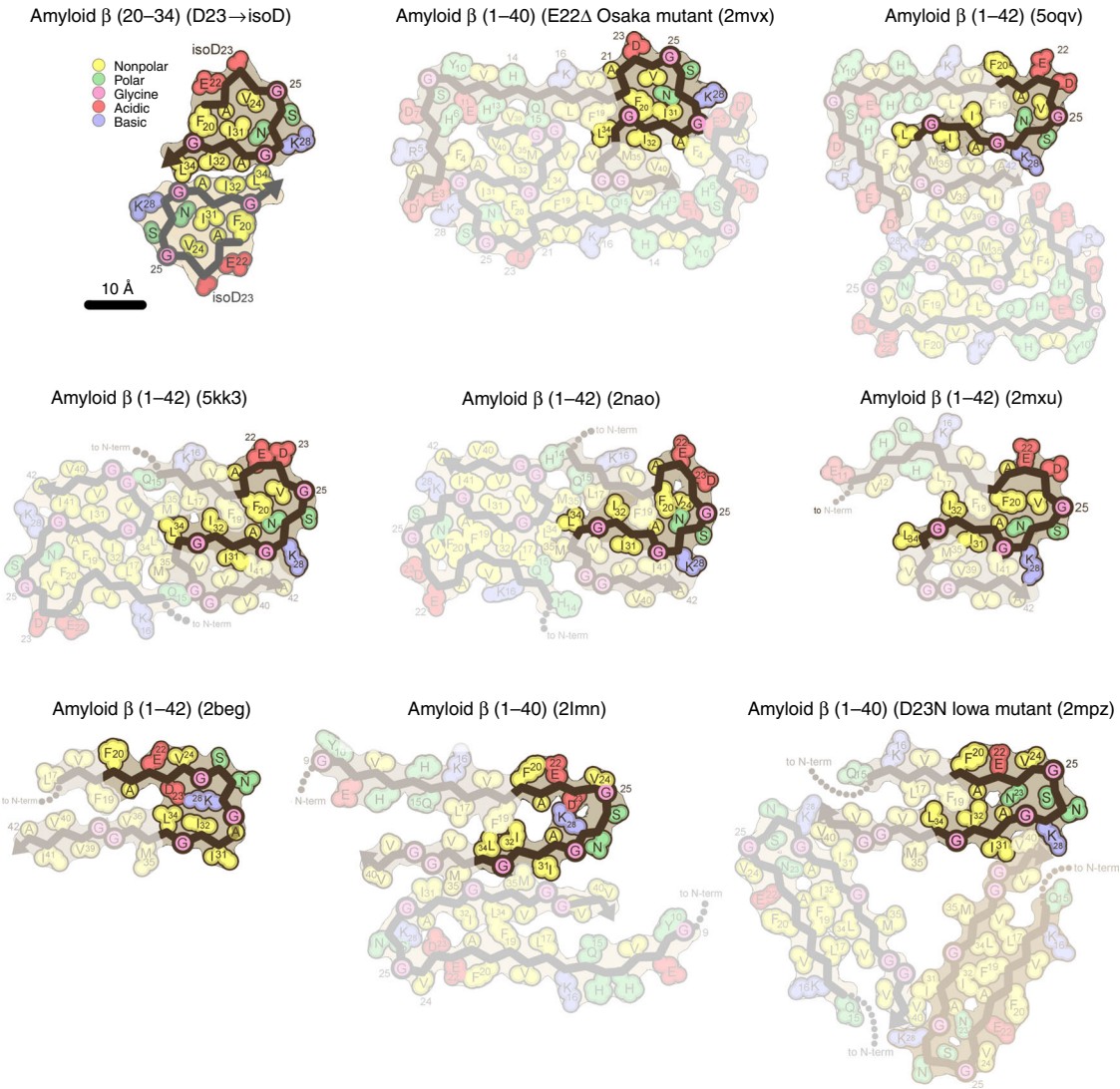

**Fig. 6** Aβ$^{20-34, isoAsp23}$ core assumes a similar fold to full-length native Aβ structures. Schematic diagrams of residues 20–34 of previously solved Aβ structures[21-26,28,29]. The structures most divergent from Aβ$^{20-34, isoAsp23}$ are shown in the bottom row. Residues are colored according to general chemical properties (legend—top left)

also provide insight into the mechanisms behind the A21G, E22G, and E22Δ hereditary mutations that introduce flexibility into the same region of the backbone. Importantly, we have also found that the only known repair pathway for L-isoAsp, the enzyme PCMT1, is unable to fully methylate and repair aggregates of Aβ$^{20-34, isoAsp23}$ in vitro, thus once the modified aggregates have formed in vivo they may be difficult to repair and clear (Supplementary Fig. 1).

Recent structures of tau isolated from AD patients have revealed distinct structural polymorphs[48]. Both the paired helical filaments and the straight filaments of tau display β-arches in their sheets, which is a feature also shared by the native and isomerized Aβ$^{20-34}$ structures (Fig. 6 and Supplementary Fig. 4). This similarity not only suggests that our structure's β-helix-like turn may be a common amyloid motif but also identifies a potential cross-seeding site between Aβ and the tau protein of AD. This discovery emphasizes the need for atomic-resolution structures of disease-associated amyloid, as these core segments are critical for structure-based drug design and protein prediction efforts[49-52]. These crystal structures can be used in conjunction with full-length cryo-EM structures to obtain a high-resolution view of the interactions mediating amyloid fiber formation[53]. High-resolution

structures are also valuable when looking at the effect PTMs may have on amyloid structure as seen here and elsewhere[54]. Therefore, the combination of increasing peptide length and high resolution makes the Aβ$^{20-34}$ and Aβ$^{20-34, isoAsp23}$ structures an important step forward for the structural characterization of amyloid proteins and their role in disease.

## Methods

**Materials**. Aβ$^{20-34}$ peptides corresponding to the human sequence were purchased from and validated by Genscript at a purity of ≥98% as the trifluoroacetic acid salt and were stored at −20 °C. Peptides were validated by electrospray ionization–mass spectrometry (ESI-MS) performed by Genscript. Aβ 1–42 was purchased from Bachem Americas, Inc. (Catalog #, H-1368).

**Aggregation of Aβ$^{20-34}$ peptides for fibril-formation rates**. Peptides were dissolved at 1.6 mM in 50 mM Tris-HCl, pH 7.6, 150 mM NaCl (TBS) with 2.5% DMSO unless otherwise designated in the figure legend. Peptides solutions were filtered through 0.22-μm cellulose acetate Costar Spin-X centrifuge tube filters (Corning Inc., product #8161). Filtered peptide solutions in a final volume of 100 μL/well in a 96-well plate (Fisherbrand, 12565501) were read at 340 nm in a Varioskan plate reader at 37 °C with continuous shaking at 1200 rpm. Readings were recorded every 15 min.

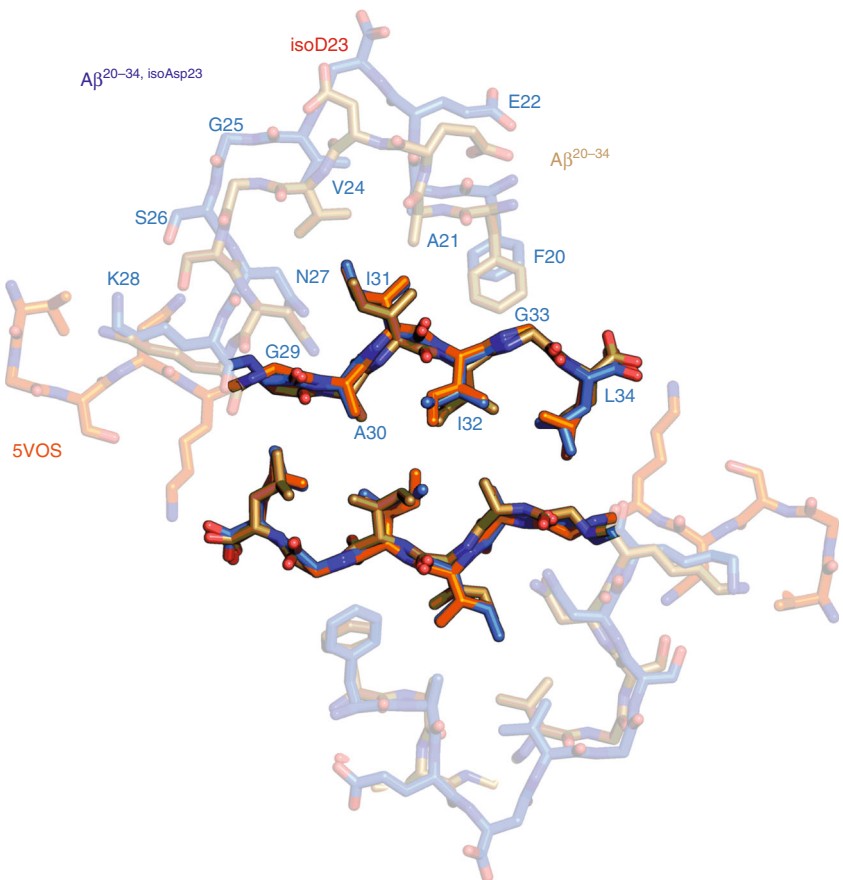

**Fig. 7** Aβ 24–34 peptide structure shares similarities with the Aβ[20–34] structures. The Aβ steric zipper structure with the lowest total atom root-mean-square deviation of all the short Aβ segment structures, PDB: 5VOS[32] (orange), is shown aligned with interface A of Aβ[20–34] (gold) and Aβ[20–34, isoAsp23] (blue)

**Seeding of Aβ[20–34] segment**. Seeds were formed shaking continuously on an acoustic resonant shaker at 37 °C at a frequency setting of 37[33,34]. Seeds of Aβ[20–34] were formed at 5 mg/mL in 50 mM Tris-HCl, pH 7.6, 150 mM NaCl (TBS) with 1% DMSO; seeds of Aβ[20–34, isoAsp23] were formed at 2.5 mg/mL in 50 mM Tris-HCl, pH 7.6, 150 mM NaCl (TBS) with 1% DMSO; and seeds of Aβ[20–34, Asp23Asn] were formed at 2.5 mg/mL in 100 mM Tris-HCl, pH 7.5, 10% isopropanol, and 200 mM sodium acetate. All seeds were diluted to 200 μM stocks and 5 μL were added to 3.2 mM Aβ[20–34] in a final volume of 100 μL. Not all wells of the unseeded 3.2 mM Aβ[20–34] condition aggregated within the time course of this assay (Fid. 1d). Solutions were read in a 96-well plate at 340 nm in a Varioskan plate reader at 37 °C with continuous shaking at 1200 rpm. Readings were recorded every 15 min.

**Synthesis and purification of native Aβ 1–40**. The syntheses of Aβ (1–40) WT and Aβ (1–40) IsoAsp23 were completed in a CEM Liberty BlueTM Microwave Peptide Synthesizer. The crude peptides were purified using an Interchim puri-Flash® 4125 Preparative Liquid Chromatography System. Details of the syntheses and purifications are available in the Supplementary Methods section (Supplementary Tables 1–3, Supplementary Figs. 6–22).

The purified Aβ (1–40) WT has an estimated purity of 93% by high-performance liquid chromatography (HPLC; Supplementary Figs. 14 and 15) and was characterized by ESI-MS via direct injection into a Q-Exactive[TM] Plus Hybrid Quadrupole-Orbitrap TM Mass Spectrometer (Supplementary Figs. 16 and 17). The calculated average mass for $C_{194}H_{295}N_{53}O_{58}S$: 4327.148 g/mol, $m/z$ calculated: $[M+3H]^{3+} = 1443.39$; $[M+4H]^{4+} = 1082.79$; $[M+5H]^{5+} = 866.44$; $[M+6H]^{6+} = 722.20$. Observed: 1443.3913; 1082.7955; 866.4374; 722.1991. The purified Aβ (1–40) IsoAsp23 has an estimated purity of 97% by HPLC (Supplementary Figs. 19 and 20) and was characterized by ESI-MS via direct injection into a Q-Exactive[TM] Plus Hybrid Quadrupole-Orbitrap TM Mass Spectrometer (Supplementary Figs. 21 and 22). The calculated average mass for $C_{194}H_{295}N_{53}O_{58}S$: 4327.148 g/mol, $m/z$ calculated: $[M+3H]^{3+} = 1443.39$; $[M+4H]^{4+} = 1082.79$; $[M+5H]^{5+} = 866.44$. Observed: 1443.3912; 1082.7959; 866.4373.

**Crystallization of the segments**. Aβ[20–34] was resuspended at a concentration of 3.2 mM in 50 mM Tris-HCl, pH 7.5, 150 mM NaCl (TBS) with 1% DMSO in a final volume of 100 μL. The peptide solution was then shaken continuously for 30 h at 1200 rpm at 37 °C. Aβ[20–34,isoAsp23] was resuspended at a concentration of 1.6 mM in 50 mM Tris-HCl, pH 7.6, 150 mM NaCl (TBS) with 1% DMSO in a final volume of 200 μL. The filtered peptide solution was then shaken for 2 days on an acoustic resonant shaker at 37 °C at a frequency setting of 37[24,25]. Four microliters of this suspension was then used to seed 196 μL of a second peptide solution (1.6 mM) as a 2% seed on the acoustic resonant shaker at 37 °C. Crystals were obtained within 48 h. The presence of crystals was verified by EM, using a standard holder, with no negative stain. Crystals of the native and isomerized segments were on average ~77 and ~71 nm in width, respectively, and were typically >2 μm in length.

**MicroED sample preparation**. Quantifoil R1.2/1.3 cryo-EM grids (Electron Microscopy Sciences, product # Q325CR1.3) were glow discharged for 30 s on either side, and 1.5 μL of a 1:1 dilution of Aβ[20–34] crystals in 50 mM Tris-HCl, pH 7.5, 150 mM NaCl (TBS) with 1% DMSO was pipetted on both sides. Twenty microliters of Aβ[20–34,isoAsp23] crystal suspensions were spun down at 5000 × g for 5 min, the supernatant was removed, and pelleted crystals were resuspended in 50 μL TBS + 0.75% (w/v) β-octyl-glucoside (VWR, P-1110), and rotated at 4 °C for 1 h. These detergent-treated crystals were then spun down a second time. Pelleted crystals were resuspended in 50 μL water. A total of 1.5 μL of the washed crystal solution was then applied on both sides of a glow discharged Quantifoil R1.2/1.3 cryo-EM grid (Electron Microscopy Sciences, product # Q325CR1.3). All grids were plunge frozen into supercooled ethane using a Vitrobot Mark 4 instrument.

**MicroED data collection and processing**. MicroED data was collected in a manner similar to previous studies[42]. Briefly, plunge-frozen grids were transferred to an FEI Talos Arctica electron microscope and diffraction data were collected using a bottom-mount CetaD 16M CMOS camera with a sensor size of 4096 × 4096 pixels, each 14 × 14 μm. Diffraction patterns were recorded by operating the detector in continuous mode with 2 × 2 pixel binning, producing datasets with frames 2048 × 2048 pixels in size. The exposure rate was set to <0.01 $e^-$/A²/s. The exposure time per frame was set at 3 s while the rotation speed was set to 0.3 deg/s resulting in a final oscillation range of 0.9 deg/exposure for the Aβ[20–34] data collection and to 0.443 deg/s resulting in a final oscillation range of 1.329 deg/exposure for the Aβ[20–34,isoAsp23] data collection. This rotation rate was optimized to allow a maximum amount of reciprocal space to be sampled before crystal decay

was observed while also slow enough to prevent overlapping diffraction spots in the diffraction images. Diffraction movies typically covered a 50–140 deg wedge of reciprocal space and were taken of crystals randomly orientated on the grid with respect to the incident beam. These crystals had a highly preferred orientation on the grid, resulting in a systematic missing cone and hence lower completeness along the $c^*$ axis; however, this did not preclude structure determination, with a high overall completeness of >80% for both structures (see Table 1).

**Structure determination.** Diffraction datasets were converted to SMV format to be compatible with the X-ray data processing software[55]. Data were indexed and integrated using XDS[56]. The parameters controlling the raster size during indexing and integration were optimized to reduce contributions by background and to exclude intensities that conform poorly to the lattice determined during indexing. The number of diffraction images used per crystal was aggressively pruned to maximize $I/\sigma$. The resulting outputs from XDS were sorted and merged in XSCALE. To produce a final merged dataset, partial datasets were selected based on their effects on the Rmerge values. In total, for the A$\beta^{20-34}$ structure, 10 partial datasets, containing 404 diffraction images, were merged to produce a final dataset with high completeness up to 1.1 Å. An ab initio solution was achieved using SHELXD[57]. In total, for the A$\beta^{20-34, \text{isoAsp23}}$ structure, 5 partial datasets, containing 159 diffraction images, were merged to produce a final dataset with high completeness up to 1.1 Å, and an ab initio solution was also achieved using SHELXD. The phases obtained from both A$\beta^{20-34}$ coordinates produced by SHELX were used to generate maps of sufficient quality for subsequent model building in Coot[58]. The resulting models were refined with Phenix[59], using electron scattering form factors, against the measured data.

**Powder diffraction sample preparation and data collection.** Designated aggregates of A$\beta$ 1–42 and A$\beta^{20-34}$ peptides were prepared in buffers as described in the figure legends. Aggregates were spun at $20,000 \times g$ for 5 min. The pellet was resuspended in water and re-spun. Pelleted fibrils were resuspended in 5 µL water and pipetted between two facing glass rods that were 2 mm apart and allowed to dry overnight at room temperature. These glass rods with ordered fibrils were secured to a brass pin and mounted for diffraction at room temperature using 1.54 Å X-rays produced by a Rigaku FRE+ rotating anode generator equipped with an HTC imaging plate. Patterns were collected at a distance of 200 mm and analyzed using the ADXV software package[60].

**SDS dissolution of aggregates.** Aggregates of A$\beta^{20-34}$, A$\beta^{20-34, \text{isoAsp23}}$, and A$\beta^{20-34, \text{Asp23Asn}}$ were all prepared in TBS, with 1%, 2.5%, and 2.5% DMSO, respectively. Both A$\beta^{20-34, \text{isoAsp23}}$ and A$\beta^{20-34, \text{Asp23Asn}}$ were prepared at a peptide concentration of 2.5 mg/mL, while A$\beta^{20-34}$ was prepared at 5 mg/mL, shaking at 1200 rpm at 25 °C. The A$\beta^{20-34}$ was diluted to 2.5 mg/mL prior to the denaturation assay. Suspensions of A$\beta^{20-34}$ aggregates were diluted 1:1 in 2, 3, 4, and 10% SDS stocks in TBS and heated for 15 min at 70 °C in a PTC-100 Peltier thermal cycler as described by Guenther et al.[54]. Measurements at 340 nm were recorded on a Nanodrop 2000 instrument. Two microliters of each solution was analyzed by EM for remaining aggregates on glow discharged Formvar/Carbon 400 mesh, Copper grids (Ted Pella, Catalog # 01754-F).

**Analysis of $S_a$ and surface $S_c$ in A$\beta^{20-34}$ structures.** The structures of A$\beta^{20-34}$ and A$\beta^{20-34, \text{isoAsp23}}$ were used to measure buried surface area ($S_a$) and ($S_c$) from an assembly consisting of two sheets generated by translational symmetry each consisting of ten stacked β-strands. $S_a$ was calculated as the average of the buried surface area per chain and the difference between the sum of the solvent accessible surface area of the two sheets and the solvent accessible surface area of the entire complex, divided by the total number of strands in both sheets using the CCP4 suite.

**Modeling modified and full-length A$\beta$ and RMSD calculations.** Residues 1–42 of A$\beta$ were modeled onto the N- and C-termini of the A$\beta^{20-34, \text{isoAsp23}}$ structure using Coot, and the resulting structures were energy minimized using the Crystallography & NMR System (CNS)[61] suite of programs.

Distance matrices for RMSD relationships between A$\beta^{20-34, \text{isoAsp23}}$ and residues 20–34 from native structures were generated in the LSQKAB program of CCP4, and resulting matrices were used to generate the tree shown in Fig. 4.

**Reporting summary.** Further information on research design is available in the Nature Research Reporting Summary linked to this article.

## Data availability
Atomic coordinates and structure factors for the A$\beta^{20-34}$ structure have been deposited in the Protein Data Bank under accession code 6OIZ. The map for this structure has been deposited in the EMDB with accession code EMD-20082. Atomic coordinates and structure factors for the A$\beta^{20-34, \text{isoAsp23}}$ structure have been deposited in the Protein Data Bank under accession code 6NB9. The map for this structure has been deposited in the EMDB with accession code EMD-0405. The source data underlying Figs. 1c, d, 2, and 4b and Supplementary Figs. 1 and 2 are provided as a Source Data file. Other data are available from the corresponding author upon reasonable request.

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

## Acknowledgements

We thank Dr. Jose Rodriguez for his advice throughout the course of this work. This work was supported in part by grants from the National Science Foundation (MCB-1714569 to S.G.C. and MCB-1616265 to D.S.E.), the National Institutes of Health (AG 054022 to D.S.E.), and the Howard Hughes Medical Institute (to D.S.E. and T.G.). R.A.W. was supported by a USPHS National Research Service Award GM007185 and a UCLA Dissertation Year Fellowship and Pauley Fellowship Award. D.R.B. was supported by a National Science Foundation Graduate Research Fellowship. L.R. was supported by a USPHS National Research Service Award 5T32GM00849. Work in the S.G.C. laboratory was also supported by funds from the Elizabeth and Thomas Plott Chair in Gerontology of the UCLA Longevity Center, a grant from the Life Extension Foundation, and funds from a UCLA Faculty Research Award. We acknowledge the use of instruments at the Electron Imaging Center for Nanomachines supported by UCLA and by instrumentation grants from NIH (1S10RR23057 and 1U24GM116792) and NSF (DBI-1338135). We thank Michael Collazo at the UCLA-DOE Macromolecular Crystallization Core Technology Center for crystallization support. This work was supported in part by NIGMS grant No. R35GM128867 and the Beckman Young Investigators (BYI) Program awarded to J.A. Rodriguez. The mass analysis of Aβ(1–40) wild-type and IsoAsp23 was supported by the National Institutes of Health under instrumentation grant 1S10OD016387–01 with the assistance of Yu Chen in the Molecular Instrumentation Center of UCLA.

## Author contributions

R.A.W. and S.G.C. designed the project and wrote the manuscript with input from all other authors. R.A.W. conducted fibril growth experiments, stability assays, and crystallization of the Aβ20–34 peptide. R.A.W. and D.R.B. performed fibril diffraction studies. Samples were prepared for MicroED and data were collected by R.A.W. and D.R.B. with advice from T.G. Data were processed by C.-T.Z. and R.A.W. and refined by R.A.W., L.R., and M.R.S. with significant contributions to data processing by D.R.B., D.C. and M.R.S. R.A.W. and M.R.S. conducted computational analysis including buried and accessible surface area, RMSD, and designed models of full-length Aβ fibrils. All authors contributed to the analyses of the structures.

## Additional information

**Competing interests:** D.S.E. is an advisor and equity shareholder in ADDRx, Inc. The other authors declare no competing interests.

