## [Peer Review File · Nature Communications]

Reviewers' comments:

Reviewer #1 (Remarks to the Author):

This manuscript presents a 1.1 Å resolution fibril structure of a fragment of the Aβ peptide comprising residues 20-34, containing an isoaspartate modification at residue 23, determined by micro-electron diffraction.

The main advance here is that the authors succeeded to determine the fibril structure of a 15 amino acid peptide by microED. Previously, only fibril structures of 11 amino acid peptides were determined by microED. Like the shorter peptides, the Aβ(20-34) peptide forms an in-register, parallel β-structure within the crystals. However, the increased length of the Aβ(20-34) peptide enables formation of a backbone conformation with two kinks at its glycine residues, allowing formation of a β-helix-like fold, while the previous microED fibril structures showed no kinks (single β-strands) or single kinks.

While it is interesting to see this high-resolution structure formed by this Aβ segment, known to form turns, the wider importance for the amyloid field is limited:

1) The relevance of studying the 15-residue segment is unclear, as it probably does not represent the full-length fibril structure. Figure 6 illustrates that the addition of 4 residues, extending Aβ(24-34) to Aβ(20-34, IsoAsp23), has an impact on the fibril structure. It would be surprising if addition of 25 further residues would not impact the structure considerably. The Aβ(20-34) structure does not particularly well match any of the already available structures of full-length Aβ (Fig. 5). It seems feasible today to aim for the fibril structure of the full-length L-isoAsp23 Aβ using cryo-EM.
2) The crystal structure was only determined for the L-isoAsp23 version of the peptide, not for the wildtype (Asp23) and Iowa mutant (Asn23). Therefore, the analysis of the effect of the L-isoAsp23 modification is limited not only by the restriction to the Aβ(20-34) segment, but also by the indirect comparison with wildtype and Iowa mutant based on models and powder diffraction data. Consequently, the discussion of the effect of the L-isoAsp23 modification is largely speculative.

Further points:

3) The statement in the last two sentences of the abstract is already substantiated by the literature on the full-length proteins (for example van Nostrand et al. J. Biol. Chem. 2001;276:32860–32866; references 11, 20)
4) The statement in the abstract “By powder diffraction we find that this structure is conserved in 27 crystals of an analogous segment containing the heritable Iowa mutation, Asp23Asn” is too strong. The powder diffraction pattern in Fig. S3 cannot serve as evidence for a common structure of Asp23Asn and isoAsp23, and a different structure of the wildtype. Another point of concern here is that the wildtype powder diffraction sample was prepared in a different buffer (different pH, salt, organic co-solvent) than the other two, which limits the informative value of the comparison of these samples.
5) Fig. 1c: Why does the Aβ(20-34) already show a considerable A340 value at t=0? Are there already aggregates at t=0? Images of the aggregates should be provided.
6) Fig. 1d: How were the fiber stocks prepared? According to the protocol of Fig. 1c (where aggregates of the wildtype are present at t=0 or do not form at all) or according to the powder diffraction protocol (where wildtype aggregates are formed in different solution conditions than Asp23Asn and isoAsp23). In either case the comparability of the dissolution data is limited. Images of the aggregates should be provided.

Reviewer #2 (Remarks to the Author):

The manuscript entitled: Structure of amyloid-β (20-34) with Alzheimer's-associated isomerization at Asp23 reveals a novel protofibril interface, presents the microED crystal structure of an Aβ

peptide (residues 20-34) harboring the post-transcriptional modification (PTM) L-isoAsp23. The structure features a novel steric interface facilitated by L-isoAsp23 which might explain the higher nucleation (and therefore, fibril aggregation) with respect to the native A β peptide. Moreover, clinical findings have identified L-isoAsp 23 in senile plaques of Alzheimer disease patient's samples. and thus might contribute to its pathogenesis.

The manuscript has broad interest, it is well written and the data supports most of its conclusions. The structure was solved using electron diffraction (microED) on nano-crystals. This is a novel technique that has harbored important progress to the field of structural studies of fibrils.

A possible drawback of this work is the fact that the authors cannot establish with certainty whether differences between the L-isoAsp 23 or native folds are the result of 1) "intrinsic" differences in protofilament packing or 2) different packing of the protofilaments in the crystals. However, the microED structure hints at a mechanism of enhanced fibril formation as a result of the presence of L-isoAsp 23.

Comments about the manuscript.

Line 114, bottom paragraph, Fig 2c is listed before Fig 2b. Figure 2c should be Fig 2b and vice versa.

Line 124, it might not be evident to the reader how this "ladder of hydrogen bonds" is formed, perhaps creating a stereo figure would help point this out.

Line 139, it would be beneficial to the reader to explain that ASA is a measure of the area exposed to solvent.

Figure 5 is not needed in the text. It could be moved to Supplementary materials.

Supplementary Fig. 4 is not easy to visualize, perhaps is better to illustrate only the C-alpha traces with colors that have better contrast and show only critical side chain residues.

Line 243. It is not clear in Supplementary Fig. 4 how this happens. Need to point out where Ser is in the figure.

Line 298, Methods section. Specify whether negative stain or cryo electron microscopy were used to identify nanocrystals.

Line 300, what size crystals were obtained?

We have now revised the paper with the constructive comments of the reviewers. A major improvement in the revised version based on their comments is the addition of a structure of the native amyloid- β (20-34) segment. This structure and its analysis have been added to Figures 3, 4, 7 and Supplementary Figures 2 and 4. In addition, two new experiments have been added to address the significance of these segments in seeding amyloid (Fig. 1d, Supplementary Fig. 1). We believe these revisions, as well as incorporation of modifications from the reviewers' others comments, significantly strengthen the manuscript. The specific changes to the manuscript are described below.

RESPONSES TO REVIEWER 1

1) Comment 1.1 (Reviewer 1, Comment 1): *The relevance of studying the 15-residue segment is unclear, as it probably does not represent the full-length fibril structure. Figure 6 illustrates that the addition of 4 residues, extending A β (24-34) to A β (20-34, IsoAsp23), has an impact on the fibril structure. It would be surprising if addition of 25 further residues would not impact the structure considerably. The A β (20-34) structure does not particularly well match any of the already available structures of full-length A β (Fig. 5). It seems feasible today to aim for the fibril structure of the full-length L-isoAsp23 A β using cryo-EM.*

2) Reply 1.1: The arrangement of the residues within both the wild-type and isomerized A β (20-34) segments correspond particularly well to the available full-length A β structures of PDB 2MVX, 2MXU, 5KK3 and 2NAO with regards to several key features. In each of these four previously solved structures and the A β (20-34) segments presented here, Phe20 is buried in a hydrophobic core, there are kinks in the polypeptide backbone about Gly25 and Gly29, and the charged residues Glu22, Asp23, and Lys28 are surface exposed. The four structures referenced above were solved by NMR methods, and we believe the conservation between these NMR studies and this crystallographic study of the features listed indicate that they consistently form a favorable protofilament core. We believe it is significant that these same features have not been observed within A β segments crystallographically before. The authors agree with the reviewer that a cryoEM structure of a full-length L-isoAsp23 A β would be informative. Because of the potentially small local changes of the L-isoAsp residue (side chain truncation of 1.5 Å and backbone extension of 1.5 Å), we chose MicroED as a starting point for investigating the L-isoAsp residue structurally, in order to ensure high resolution information.

We have now revised the manuscript text to make these points clearer.

3) Comment 1.2: *The crystal structure was only determined for the L-isoAsp23 version of the peptide, not for the wildtype (Asp23) and Iowa mutant (Asn23). Therefore, the analysis of the effect of the L-isoAsp23 modification is limited not only by the restriction to the A β (20-34) segment, but also by the indirect comparison with wildtype and Iowa mutant based on models and powder diffraction data. Consequently, the discussion of the effect of the L-isoAsp23 modification is largely speculative.*

4) Reply 1.2: We have now obtained and added the structure of the wildtype A β (20-34) segment to the manuscript in order to highlight the structural differences between the unmodified and modified forms. The Iowa mutant segment has remained recalcitrant to crystallization, but work is ongoing. As requested, we have now limited our speculation on the structure of the Iowa mutant to Fig. 4 within the paper.

- 5) **Comment 1.3:** *The statement in the last two sentences of the abstract is already substantiated by the literature on the full-length proteins (for example van Nostrand et al. J. Biol. Chem. 2001;276:32860–32866; references 11, 20)*
- 6) **Reply 1.3:** Here the reviewer refers to the statements of the abstract: *“In correspondence with its early onset phenotype, Asp23Asn accelerates aggregation of A β 20-34, as does the L-isoAsp23 modification. The enhanced amyloidogenicity of these modified A β segments may reduce the concentration required to achieve nucleation and contribute to the pathogenesis of AD.”* The authors hoped to emphasize that the L-isoAsp modification may act pathogenically in a similar manner to the Iowa mutant, by accelerating the nucleation step of amyloid formation. We have revised the last sentence with the aim of clarifying these comparisons of native, modified, and mutant: *“These structures suggest that the enhanced amyloidogenicity of the modified A β segments may also reduce the concentration required to achieve nucleation and therefore help spur the pathogenesis of AD.”*
- 7) **Comment 1.4:** *The statement in the abstract “By powder diffraction we find that this structure is conserved in 27 crystals of an analogous segment containing the heritable Iowa mutation, Asp23Asn” is too strong. The powder diffraction pattern in Fig. S3 cannot serve as evidence for a common structure of Asp23Asn and isoAsp23, and a different structure of the wildtype. Another point of concern here is that the wildtype powder diffraction sample was prepared in a different buffer (different pH, salt, organic co-solvent) than the other two, which limits the informative value of the comparison of these samples.*
- 8) **Reply 1.4:** The reviewer makes an excellent observation of possible differences in the powder diffraction patterns due to discrepancies in the fiber preparations. We have removed these powder diffraction experiments and replaced them with the ones now shown in Fig. 4, in which all fibers were prepared in the same buffer (Tris-HCl), salt (NaCl), and co-solvent (DMSO), with the only variable being a slightly higher concentration of DMSO needed for solubilization of the Iowa mutant segment (1% vs. 5%). To address the reviewer's concerns about our speculation on the Iowa mutant structure, we have also now included a seeding experiment of the wild-type segment with the three separate segments (Fig. 1d). The result shows that all three segments can greatly accelerate aggregate formation, which is a further indication that the Iowa mutant likely retains strong structural similarity with the wild type segment. We also performed powder diffraction of the resultant aggregates, in which there was remarkable overlap in the reflections from all seeded and unseeded conditions. Thus, it is unlikely that the Iowa mutant accelerated wild-type aggregation is templated by a completely unique structure (Supplementary Fig. 2a).
- 9) **Comment 1.5:** *Fig. 1c: Why does the A β (20-34) already show a considerable A340 value at t=0? Are there already aggregates at t=0? Images of the aggregates should be provided.*
- 10) **Reply 1.5:** The black line shown in Fig. 1c is the buffer alone condition and has A340 values of approximately 0.135. Thus, the signal seen at t = 0 for these assays is background absorbance. We have now labeled the black line as a buffer control in the figure, as well as provided pictures of the final aggregates.

- 11) Comment 1.6:** *Fig. 1d: How were the fiber stocks prepared? According to the protocol of Fig. 1c (where aggregates of the wildtype are present at t=0 or do not form at all) or according to the powder diffraction protocol (where wildtype aggregates are formed in different solution conditions than Asp23Asn and isoAsp23). In either case the comparability of the dissolution data is limited. Images of the aggregates should be provided.*
- 12) Reply 1.6:** We have since repeated the experiment in order to provide samples for EM imaging of remaining aggregates. The SDS dissolution results were reproducible between the two experiments. All of the aggregates were prepared in TBS, with differing amounts of DMSO. The new SDS-denaturation results along with images of the aggregates are now in Fig. 2.

RESPONSES TO REVIEWER 2

- 13) Comment 2.1 (Reviewer 2, Comment 1):** *A possible drawback of this work is the fact that the authors cannot establish with certainty whether differences between the L-isoAsp 23 or native folds are the result of 1) “intrinsic” differences in protofilament packing or 2) different packing of the protofilaments in the crystals.*
- 14) Reply 2.1:** This comment is addressed by the addition of the wild-type A β 20-34 structure. The crystals for this structure were obtained in an almost identical buffer solution (50 mM Tris-HCl pH 7.5 v. 7.6), thus lowering the possibilities of generating differences in crystal packing by differing buffer conditions. We see that both interfaces are maintained, but that the novel L-isoAsp-mediated interface significantly differs from that of the wild-type.
- 15) Comment 2.2:** *Line 114, bottom paragraph, Fig 2c is listed before Fig 2b. Figure 2c should be Fig 2b and vice versa.*
- 16) Reply 2.2:** This has been addressed.
- 17) Comment 2.3:** *Line 124, it might not be evident to the reader how this “ladder of hydrogen bonds” is formed, perhaps creating a stereo figure would help point this out.*
- 18) Reply 2.3:** We have now added a panel (Supplementary Fig. 3b) to highlight the ladder of H-bonds.
- 19) Comment 2.4:** *Line 139, it would be beneficial to the reader to explain that ASA is a measure of the area exposed to solvent.*
- 20) Reply 2.4:** With the wild type structure, the ASA discussion is no longer as relevant and we have deleted it.
- 21) Comment 2.5:** *Figure 5 is not needed in the text. It could be moved to Supplementary materials. Supplementary Fig. 4 is not easy to visualize, perhaps is better to illustrate only the C-alpha traces with colors that have better contrast and show only critical side chain residues.*
- 22) Reply 2.5:** The authors believe an important feature of these segment structures is the similarity to available full-length A β structures. To emphasize this, Fig. 5 has been left in the text (**now Fig. 6**), in order to emphasize the following points within the text: Phe20 is

buried in a hydrophobic core, kinks are present in the polypeptide backbone about Gly25 and Gly29, and the charged residues Glu22, Asp23, and Lys28 are surface exposed in the structures with this lowest RMSD with our structures. Supplementary Fig. 4 (**now Supplementary Fig. 5**) has been altered from comparing the L-isoAsp segment structure with the available structures, to comparing the wild-type segment structure to the available structures. This has been changed to a color (gold) that makes it easier to visualize comparisons against the residues of the other structure. We feel it is important to keep the side chains, particularly to show the similarities in placement of the normal Asp residue. We have now made these side chains transparent to show better contrast as the reviewer suggests.

23) Comment 2.6: *Line 243. It is not clear in **Supplementary Fig. 4** how this happens. Need to point out where Ser is in the figure.*

24)Reply 2.6: The discussion of the Ser has been removed from the text now that there are direct comparisons that can be made with the wild-type segment structure.

25) Comment 2.7: *Line 298, Methods section. Specify whether negative stain or cryo electron microscopy were used to identified nanocrystals.*

26)Reply 2.7: We have now added a statement in the methods to clarify the conditions used.

27) Comment 2.8: *Line 300, what size crystals were obtained?*

28) Reply 2.8: This information has been added to the methods section.

REVIEWERS' COMMENTS:

Reviewer #1 (Remarks to the Author):

The manuscript improved considerably thanks to the inclusion of the wildtype structure. This now allows to draw conclusions on the structural consequences of the L-isoAsp modification. My comments have been satisfactorily addressed.